# One-Leg Stance Postural Sway Is Not Benefited by Bicycle Motocross Practice in Elite Riders

**DOI:** 10.3390/jfmk8010025

**Published:** 2023-02-16

**Authors:** Carlos Albaladejo-García, Francisco J. Moreno, Fernando García-Aguilar, Carla Caballero

**Affiliations:** Sports Research Centre, Department of Sport Sciences, Miguel Hernández University, 03202 Elche, Spain

**Keywords:** postural sway, non-linear tools, laterality, cycling, sport

## Abstract

Balance has been positioned as an important performance skill in sport. Differences in postural control have been found between levels of expertise. However, this statement remains unanswered in some cyclic sports. This work aimed to describe the one-leg balance performance of a sample of elite BMX riders—racing and freestyle—compared to a control group formed by recreational athletes. The center of pressure (COP) of nineteen international BMX riders (freestyle, n = 7; racing, n = 12) and twenty physically active adults was analyzed in a 30-s one-leg stance test on both legs. COP dispersion and velocity variables were analyzed. Non-linear dynamics of postural sway were evaluated through Fuzzy Entropy and Detrended Fluctuation Analysis. BMX athletes did not show differences between legs in any of the variables. The control group did show differences between the dominant and non-dominant leg in the magnitude of variability of the COP in the mediolateral axis. Group comparison revealed non-significant differences. International BMX athletes did not show better balance parameters than the control group in a one-leg stance balance task. The adaptations derived from BMX practice do not have a significant impact in one-leg stance balance performance.

## 1. Introduction

Balance is defined as the ability to maintain the center of gravity of the individual on the base of support [1,2]. For Barone et al. [3], balance is an indispensable skill within motor behavior, which is achieved through muscular synergies that minimize the displacement of the Center of Pressure (COP). Previous studies have proposed that a good development of balance has particular importance in the sports arena because it is essential to perform fundamental motor skills such as jumping, throwing, kicking, or hitting [1]. Therefore, balance has been deeply studied in sport sciences, frequently using analyses of COP dispersion [4] but the optimal selections of discriminative sway parameters are still unclear [5]. In recent years, some authors have proposed the analysis of balance dynamics through non-linear tools as a complementary analysis to assess the postural sway dynamics [6]. These tools allow the quantification of the motor behavior changes over time [4,7], giving information about the dynamic characteristics of sway patterns, which seem to be related to the ability to perform movement adjustments [8,9]. These measures have been applied to assess the COP dynamic, being the Detrending Fluctuations Analysis (DFA) and entropy the most widely used. Higher complexity of COP fluctuations, measured by DFA, has been related to the flexibility shown by younger women compared to older women in executing motor adjustments [10,11,12]. In addition, it has also been applied to relate postural balance to performance in different sporting skills [4,13,14].

Previous studies have supported the idea that the experience in sports such as soccer, gymnastics, shooting sports, or golf can induce balance adaptations characterized by a higher ability to perform postural adjustments during balance [15,16,17,18]. For example, Jadczak et al. [16] reported that the higher the competitive level of soccer players, the better their postural control, making them more effective in the actions specific to their sport. A common belief is that high-level sports training, due to highly competitive demands, causes better sensory organization and motor performance [19]. Paillard [20], in a review on the role of balance in sports performance, found that the practice of physical exercise could induce structural and functional adaptations in the postural control system. Nevertheless, previous studies have found differences in balance performance between athletes of different sports modalities, suggesting that only certain sports would benefit from the development of this skill among their practitioners [21]. This is the case of gymnasts, whose performance is based on the execution of different technical skills that require great postural mastery, both in static and dynamic situations [15]. Similarly, soccer players have shown better one-leg balance than other athletes such as basketball players or swimmers, possibly because the player maintains the balance with one leg while the ball is kicked with the other leg [2,22].

Balance has been positioned as a key performance skill in other sports such as cycling, being conditioned by the presence of a vehicle [23,24]. However, few studies have analyzed balance in on-road, off-road, or BMX cyclists. It has been suggested that mountain bikers are exposed to more balance-demanding situations, in which the somato-kinesthetic information is more relevant than in other specialties such as road cycling, in which visual control was predominant [25]. Another study reported a decrease in balance in a group of off-road cyclists after their racing season, an effect that was not found in on-road cyclists [24]. The authors argued that the terrain vibrations may have caused modifications in the rider’s vestibular and somatosensory system. Regarding BMX cycling, the role of balance has a special function due to the continuous technical executions that athletes must perform both on and off the ground. As mentioned, balance ability is often sport specific, although previous studies have suggested that it can also be reflected in basic actions such as balance on one leg [15]. To our knowledge, the information about the relationship between balance and competitive performance in BMX cycling is still limited. There are no data on whether BMX riders develop greater balance ability compared to other athletes. Knowing an athlete’s balance ability could help us to categorize and understand the athlete’s level of adaptation.

Similarly, it is not known whether BMX practice leads to interlimb balance asymmetries. Cyclic sports, such as swimming or running, did not show a marked asymmetry in the use of one leg or the other [21,26]. Classic on-road cycling modalities are also considered symmetric sports [27]; however, BMX cyclists perform both cyclic and acyclic actions (e.g., static and dynamic balance stances, acrobatics, spins, and jumps). In this regard, BMX riders’ balance can be conditioned by the use of their dominant leg, identified as the one that is placed in the most forward position, which is especially involved in stabilizing the bike during jumps and skills. It is still not known if the leg identified as dominant manifests higher balance levels against the contralateral one, showing functional specialization characteristics related to the practice of this sport. Knowing whether there is a specialized dominant leg could help us to detect asymmetries between legs, and it could be useful to optimize training and prevent injuries [28].

In this work, the balance of a sample of international BMX riders—racing and freestyle—will be evaluated in a one-leg support situation to describe their balance capacity in comparison with a control group formed by physically active people. It is hypothesized that the sample of athletes will present better balance parameters than the control group, shown through lower COP dispersion and speed. In addition, it is not expected to find interlimb balance asymmetries due to the cyclic participation of both legs in this sport.

## 2. Materials and Methods

### 2.1. Participants

The sample consisted of two groups, a group of BMX riders (n = 19, 7 women) and a control group (n = 20, 8 women). The BMX group was made up of 7 freestyle and 12 racing riders (age: 21.9 ± 4.4 years, body mass: 68.6 ± 12.1. kg, height: 170.6 ± 9.9 cm). Racing riders’ objective is to be the fastest to complete a predefined circuit, while freestyle BMX is based on bike acrobatics (e.g., balancing, spins and jumps). The athlete group represented the BMX Spanish national team and competed in international tournaments, having 10 (± 3) years of experience. They trained six days per week and had an annual competition volume of approximately 30–40 national and international competitions. The control group consisted of 20 physically active adults with no experience in BMX nor in sports involving balance in the performance (e.g., gymnastics, dance or surfing) (age: 23.9 ± 3.6 years, body mass: 68.4 ± 7.6 kg, height: 173.4 ± 6.9 cm). All participants were informed in advance of the procedures and the objectives of the research. Written informed consent was obtained from each participant before testing. The experimental procedures used in this study were in accordance with the Ethics Committee of the University Miguel Hernandez (code number: DPS.JSM.02.18).

### 2.2. Procedures

The participants performed a one-leg balance test. This test was chosen for two reasons: (1) so that both the BMX athletes and the control group can execute it; (2) to check if a general balance test can provide useful information, facilitating its use in field tests. Each participant performed four trials in a laboratory setting, two trials with the dominant leg and the other two with the non-dominant one. Then, the best trial of each leg was selected, considering this one the trials with lower bivariate variable error. This minimized the effect of fatigue and learning and ensured that the results reflected actual performance on the task. Leg order was counterbalanced between participants. Each trial lasted 40 s with a 30 s recovery time. Participants were instructed to stand as upright as possible to avoid displacement of their COP. Figure 1 shows the position adopted by the participants during the task: barefoot, the leg that was not on the ground was placed close to the other above the ankle, arms crossed on the chest, eyes open, and looking straight ahead [29]. If these conditions were not met, the trial was considered not valid and had to be repeated.

A Kistler model 9287CA (Zurich, Switzerland) triaxial dynamometric platform was used to record the COP measurements, which recorded the ground reaction forces at a frequency of 1000 Hz.

### 2.3. Data Analysis and Reduction

COP time series were previously down-sampled from 1000 to 20 Hz following previous suggestions to use sampling frequencies close to the COP dynamics [30], and reduce the risk of signal oversampling, which could possibly lead to artificial collinearities that could affect the variability data [31]. The first 10 s of each trial were discarded to avoid non-stationarity related to trial initiation [32]. The time series length was 600 data points.

The standard deviation (SD) and mean velocity (MV) of the COP displacement were calculated in the anteroposterior (A-P) and mediolateral (M-L) axes. The mean velocity magnitude (MVM) and the bivariate variable error (BVE) were also calculated. Fuzzy entropy (FuzzyEn) and DFA were calculated in the A-P and M-L axes to assess the complexity of the COP variability. Complexity refers to the structure of variability, i.e., how the fluctuations of the COP evolve over time [30]. Different methods of complexity can be analyzed depending on the method used. FuzzyEn values indicate the degree of irregularity in a signal. This tool has been shown to be more consistent in relative terms, less dependent on data length, free parameter selection and more resistant to noise than other entropy measures (e.g., sample entropy) [33]. This measure computes the repeatability of vectors of length m and m + 1 that repeat within a tolerance range of r of the standard deviation of the time series. Higher FuzzyEn values indicate greater irregularity in the signal time domain, whereas lower FuzzyEn values indicate greater regularity. To calculate this measure, the following parameter values were used: vector length, m = 2; tolerance window, r = 0.2 × SD; and gradient, n = 2 [33,34]. On the other hand, DFA evaluates the presence of long-term correlations within a time series using a parameter known as the scaling index α [35]. The α value identifies the extent to which proceeding data are dependent on previous outcomes [36] and it has been related to the complexity of the time series data [37]. The trial with the lowest BVE results in any leg was selected for the analysis.

Different leg dominance criteria were followed in order to discuss the most sensitive one when identifying between-groups and between-legs performance. Two criteria were followed to determine leg dominance: for all the participants the criteria applied was their preferred leg for kicking a ball [38,39]. For the group of BMX athletes only, a second criterion was applied according to the leg placed in the forward position during jumps and tricks. Additionally, group differences were analyzed in the leg that showed better balance (i.e., lower BVE), despite the participants’ preference.

### 2.4. Statistical Analysis

A sensitivity power analysis for the independent means *t*-test was performed to find out the expected effect size as a function of the study sample size. The software G*Power 3.1 (v3.1, University of Düsseldorf, Germany) was used with the following parameters: α = 0.05; 1-β = 0.80; n1 = 19; n2 = 20. This analysis showed that the effect size required for the changes to be significant with this sample would have to be d = 0.81. IBM SPSS Statistics package v26.0 was used for the rest of the analysis. The normality of the variables was assessed using the Shapiro–Wilk test. Pearson product moment correlation coefficients were calculated to assess relationships between performance variables (SD, BVE, MV, and MVM) and complexity measures (FuzzyEn and DFA). A paired *t*-test was performed to study the possible significant differences according to leg dominance within each group. An independent measures *t*-test was performed to compare the differences between groups. Because multiple balance parameters were used to assess postural control performance in the balance task, statistical significance was adjusted following Bonferroni criteria. Thus, statistical significance was set at *p* < 0.002 for correlations and *p* < 0.0016 for the *t*-test.

## 3. Results

A preliminary analysis comparing BMX freestyle and BMX racing riders was carried out. Since no differences were found between these groups, further analyses were carried out considering the athletes as a single group (n = 19). Table 1 shows the results of the correlational analysis performed to determine the relationship between COP irregularity (FuzzyEn) and autocorrelation (DFA) with COP variability (SD, BVE) and velocity (MV, MVM). Overall, dispersion variables did not correlate with COP complexity. Mean velocity COP excursion values did correlate with the two complexity measures (FuzzyEn and DFA) in the A-P axis, and with FuzzyEn in the M-L axis. Generally, participants who showed higher velocity in the COP excursion showed higher complexity.

Table 2 shows the differences between groups and between legs according to the different preferred leg criteria. BMX athletes showed no differences between legs in any of the variables for either of the two criteria. For the group control, the right leg was dominant for 95% of the participants (19/20). However, for only 40% of the sample (8/20) it was the best performing leg. While in the BMX group, the right leg was dominant for 89% of the participants (17/19), and for only 47% (9/19) it was the best performing leg. In 63% of the participants from the BMX group (12/19), the forward leg on the bicycle coincided with the leg chosen to kick a ball. Participants in the control group did show differences between the dominant and non-dominant leg in the magnitude of variability (SD) of the COP in the M-Laxis (t = 3.739, *p* = 0.001). Group comparison revealed that the BMX group showed a greater magnitude of COP variability than the control group in the SD in M-L axis (t = 3.512, *p* = 0.001), but only with BMX-Leg dominance criteria. BMX group presented higher COP magnitude and COP velocity in the rest of the variables. However, these measures, as well as non-linear tools, showed non-significant differences between groups.

Considering that the participants did not reach the best performance in the balance task with their preferred leg, an additional comparison between groups was made according to the leg with lower COP dispersion (lower BVE). BMX athletes and the control group presented a similar performance in all variables since differences between groups were non-significant (Figure 2).

## 4. Discussion

This work aimed to explore the balance of BMX riders in a one-leg support situation. The main hypothesis was that the group of BMX athletes would present better balance parameters than the control group, but it would not be mediated by the use of the dominant leg due to the cyclic participation of both legs during the sports practice.

The results did not entirely confirm the hypothesis as BMX athletes have shown worse balance performance, showing a significantly greater COP dispersion (SD M-L) than the control group in the non-dominant leg when BMX-leg dominance criteria were used. These data seem to indicate a lower ability to maintain COP on the base of support as expected in the BMX group. It cannot be concluded that BMX training did not benefit riders for increased balance ability. However, the results suggest that the experience of training on the bicycle does not have a particular benefit on the one-leg balance performance compared to regular physical practice. Previous studies have found that in sports in which technical executions are performed with one leg while the other leg stabilizes the action, such as soccer players, athletes showed better balance parameters than athletes of other modalities [2,3]. The researchers proposed that the more opportunities athletes have to support their body weight with one leg could facilitate an increase in the performance of this skill. In this regard, balance manifested in a one-leg stance may not be a factor that characterizes a population of elite BMX athletes.

Non-linear measurements of COP fluctuations have been implemented to explore the dynamics of balance movements. The analysis did not reveal any potential difference between groups, nor between legs, in the non-linear variables. Therefore, it cannot be stated that BMX practice leads to better COP modulation through postural control adaptation strategies in a single-leg stance task. The correlational analysis showed that the information provided by non-linear analysis is not clearly related to that provided by traditional dispersion variables such as SD or BVE. Non-linear measurements would provide complementary information about the complexity of the postural sway, related to the way in which movement adjustments occur during balance tasks, rather than the amount of displacement. Moreover, previous studies have proposed that a nonlinear approach would help to reveal the true state of the postural balance control system [4,40]. Specifically, non-linear parameters such as FuzzyEn or DFA have been applied to analyze how motor behavior changes over time, and they have been linked to some potential underlying mechanisms related to a higher ability to perform motion adjustments [4,8,9,40]. Correlational analysis did show some relationship between non-linear measurements and mean velocity of the COP. Barbado et al. [7] suggested that the MV is a representative measure of the number of corrections made by the participant during the task.

Previous studies have supported that individuals with more complex COP excursion, related to less regular (high entropy) and auto-correlated (low DFA) values, showed better performance in balance tasks [13,14]. Caballero et al. [14] even reported that handball players who presented a better balance and more complex COP excursion also exhibited higher accuracy and velocity in throwing, suggesting that greater variability in the movement would provide more resources to achieve better motor performance. Nevertheless, given the low significance of the results obtained in this study, conclusions about the relationship between the ability to execute body adjustments and complexity should be treated with caution.

When comparing the dominant and non-dominant leg, no statistically significant differences were found in the BMX group considering the kicking-leg dominance criteria. Similarly, the forward leg dominance criteria did not show significantly different values between legs in BMX athletes. These data agree with other studies that have reported the absence of asymmetry in balance between the dominant and non-dominant leg in cyclic and acyclic sports [1,2,3,21,26]. Although BMX riders perform both symmetric (i.e., pedaling) and asymmetric actions (i.e., jumping and balancing) [27], it seems that symmetric actions prevail over asymmetric actions, since the experienced BMX riders measured in this study have shown similar performance in both extremities in a one-leg stance. However, little is known about how balance asymmetries are manifested over the bicycle. In contrast, rhythmic gymnastics athletes have shown asymmetries [41] possibly because they use the same supporting leg to execute different technical elements such as balances and turns. In the present study, the control group showed asymmetry in one-leg stance, indicating significantly lower balance performance with the dominant leg compared to the non-dominant leg. Controversial results can also be found in previous sport-related experiments such as soccer [3,42,43,44]. Soccer players prefer to kick the ball with their more skilled (dominant) leg, while their non-dominant leg stabilizes the execution to achieve greater accuracy. Therefore, the non-dominant leg would present better balance parameters, as it would be highly specialized in these actions, which are continuously repeated. Although BMX riders perform technical actions by placing the same leg in the most forward position to stabilize the bicycle during jumps and tricks, this has not led to a significant asymmetry in balance performance. Perhaps differences can be found in this population in other skills, such as unilateral jumping or force application [27].

Results found between freestyle and racing modalities did not differ in their mean values, so it is not possible to conclude that the data presented could be derived from the different modalities of the BMX group participants. The similarity between the characteristics of both modalities may imply that there are no differences in the one-leg stance balance of the athletes. Additionally, it should be noted that the participants of both BMX disciplines who participated in the study are at the highest competitive level (national team), and followed similar training methodologies, minimizing the possible training differences. Nevertheless, other parameters could show differences between these modalities, such as force production, power output, torque, or kinematics [45,46,47].

Some limitations of the study should be noted. Firstly, the limited sample size. Given the number of members of the BMX national team, it was not possible to enlarge the sample size, and thus, it made it difficult to adopt more solid conclusions. Secondly, regarding the control group, its characteristics cannot be disregarded as a possible confounding factor. Even though participants with experience in sports with high balance demands were excluded, the participants in the control group practiced different types of recreational sports, so the implications of their background in the single-leg stance tested in the present study cannot be precisely evaluated. Thirdly, the no-specificity of the test can be considered another limitation. In addition, the one-leg static balance test used in the present study may not have been very demanding for any of the groups, so that test difficulty may be considered a limitation in the findings. However, this test was chosen for the comparison between groups because both the BMX athletes and the control group can easily execute it and to check if a general balance test can provide useful information. It is encouraged to use more complex tests in future research (e.g., open vs. closed eyes one-leg stances, and dynamic vs. static balance tests).

## 5. Conclusions

International BMX athletes did not show better balance parameters than a control group of physically active people in a one-leg stance balance task. No significant differences were found between racing and freestyle modalities in any of the study measures. Similarly, no significant differences were found between legs in the BMX sample, assuming symmetrical performance in one-leg balance tasks. The specificity in the adaptations derived from the use of a vehicle has not shown that these athletes acquire better balance levels because of their sporting experience. Future work could be directed at exploring complementary analysis and comparisons between other cycling modalities or using different tests to discriminate the effect on balance.

## Figures and Tables

**Figure 1 jfmk-08-00025-f001:**
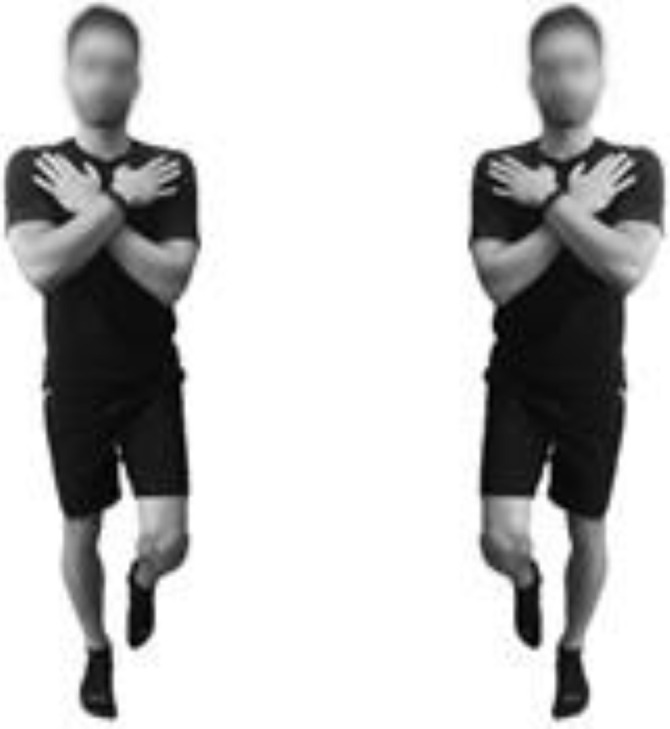
One-leg test position.

**Figure 2 jfmk-08-00025-f002:**
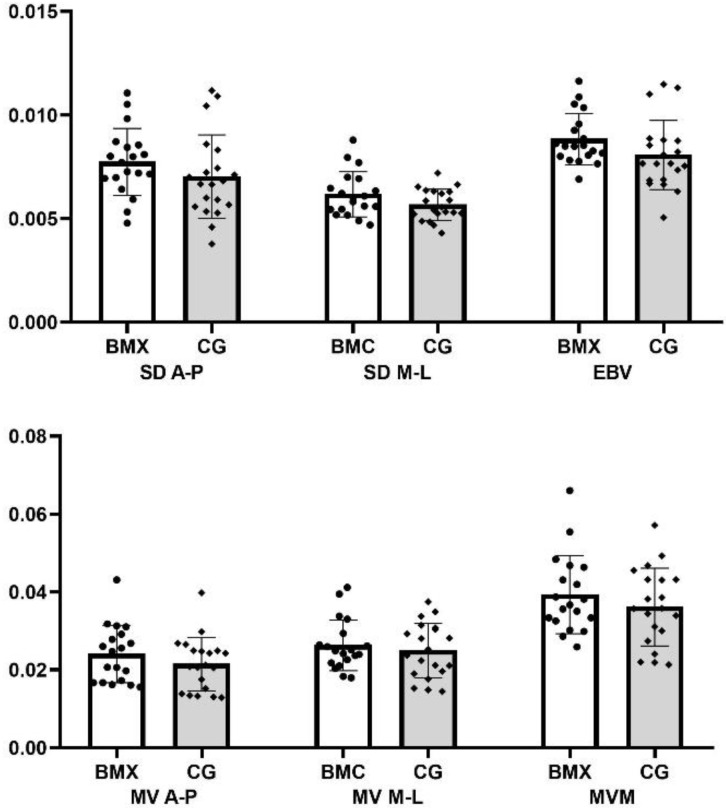
Comparison between groups using the results of the leg that showed better performance during the balance task (lower BVE). Units of center of pressure (COP) measures are as follows: cm (SD, BVE); cm/s (MV, MVM).

**Table 1 jfmk-08-00025-t001:** Correlation analysis between non-linear tools and traditional variables in the leg with better balance (lower EBV) (this analysis included BMX athletes and controls; n = 39).

		SD		BVE	MV		MVM
		A-P	M-L		A-P	M-L	
FuzzyEn	A-P	−0.31	0.16	−0.15	0.73 *	0.61 *	0.72 *
	M-L	−0.20	−0.09	−0.18	0.40 *	0.77 *	0.65 *
DFA	A-P	−0.05	−0.32 *	−0.15	−0.72 *	−0.50 *	−0.66 *
	M-L	0.15	0.23	0.19	0.02	−0.25	−0.14

* Significant correlation (* *p* < 0.002).

**Table 2 jfmk-08-00025-t002:** Means and standard deviations of SD, MV, MVM, BVE, FuzzyEn and DFA, following the criteria of the preferred leg to kick a ball (Kick-Leg) and the forward leg to perform the skills on the bicycle (BMX-Leg).

		BMX Group				Control Group	
		Kick-Leg Dominance Criteria		BMX-Leg Dominance Criteria		Kick-Leg Dominance Criteria	
		Dom.	No-Dom.	Dom.	No-Dom.	Dom.	No-Dom.
SD (cm)	A-P	0.77 ± 0.16	0.79 ± 0.25	0.80 ± 0.18	0.76 ± 0.23	0.70 ± 0.20	0.67 ± 0.14
	M-L	0.62 ± 0.11	0.63 ± 0.13	0.60 ± 0.09	0.65 ± 0.14 ^#^	0.57 ± 0.08 *	0.52 ± 0.08
MV (cm/s)	A-P	2.41 ± 0.74	2.28 ± 0.65	2.29 ± 0.62	2.39 ± 0.76	2.14 ± 0.69	1.82 ± 0.53
	M-L	2.63 ± 0.65	2.89 ± 0.91	2.62 ± 0.55	2.90 ± 0.97	2.49 ± 0.70	2.29 ± 0.64
MVM (cm/s)		3.93 ± 1.01	4.03 ± 1.12	3.83 ± 0.83	4.12 ± 1.24	3.61 ± 1.00	3.22 ± 0.86
BVE (cm)		0.88 ± 0.12	0.9 ± 0.23	0.89 ± 0.15	0.89 ± 0.22	0.81 ± 0.17	0.75 ± 0.12
FuzzyEn	A-P	0.54 ± 0.16	0.51 ± 0.15	0.50 ± 0.17	0.54 ± 0.14	0.54 ± 0.19	0.48 ± 0.14
	M-L	0.70 ± 0.12	0.74 ± 0.14	0.72 ± 0.10	0.73 ± 0.16	0.72 ± 0.16	0.71 ± 0.13
DFA	A-P	0.95 ± 0.21	1.03 ± 0.21	0.99 ± 0.20	0.99 ± 0.22	1.02 ± 0.17	1.12 ± 0.17
	M-L	0.90 ± 0.15	0.91 ± 0.12	0.90 ± 0.09	0.91 ± 0.17	0.90 ± 0.14	0.86 ± 0.16

Units of center of pressure (COP) measures are as follows: cm (SD, BVE); cm/s (MV, MVM). Dom: dominant leg; No-Dom: non-dominant leg. * Significant differences with the contralateral leg (* *p* < 0.0016). # Significant differences with control group (^#^ *p* < 0.0016).

## Data Availability

The databases used for the present study can be consulted at the following link: https://doi.org/10.6084/m9.figshare.22069103.v1.

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
