# Peer review of "One-Leg Stance Postural Sway Is Not Benefited by Bicycle Motocross Practice in Elite Riders"

_jfmk, 2023, doi:10.3390/jfmk8010025_

Round 1

Author Response

The authors would like to thank you for your advice and recommendations, which in our view have contributed to improve the quality of the paper. The authors’ responses to each comment (in bold) are presented below, and changes in the manuscript are presented in red.

Comments from Reviewer 1:

The main aim of the manuscript was to investigate whether the influence of bicycle motocross impacts balance performance (single-leg) compared to physically active adults. Elite riders from either freestyle or racing environments were used in this study. Leg dominance was included as a factor and center-of-pressure magnitude and structure variables were used to determine whether differences in postural control exist between the two population groups. Overall, no group differences were found and the general conclusion suggests that BMX practice did not have a significant effect on balance performance.

Major issues

  1. Overall, the introduction is well organized and contains appropriate rationale for the potential connection of a unique experience (BMX riding) with postural control changes. While, previous research has a little stronger connection with the idea that sport/activity experience may lead to either interlimb asymmetrical balance differences or control group difference. The authors do make an attempt to justify how BMX riding may contribute to interlimb differences based on pedaling, jumping activities involved within the sport; however, in my mind it is hard to fully comprehend how these very distinct activities fully related to balance performance given the different demands. For example, pedaling (either seated or standing) forces on force development.

Authors: We thank the reviewers for their comments. The paragraph has been reworded in lines 81-88 to emphasize why asymmetries are unlikely to be found, but it still serves an interesting purpose for the present study. In order to check the edition, please, see below:

Lines 83-90: “Similarly, it is not known whether BMX practice leads to interlimb balance asymmetries. Cyclic sports, such as swimming or running, did not show a marked asymmetry in the use of one leg or the other (Leinen et al., 2019; Samadi et al., 2009). Classic on-road cycling modalities are also considered symmetric sports (Maloney, 2019); however, BMX cyclists perform both cyclic and acyclic actions (e.g., static and dynamic balance stances, acrobatics, spins, jumps, etc.). In this regard, BMX riders’ balance can be conditioned by the use of their dominant leg, identified as the one that is placed in the most forward position”.

  1. Background information and rationale needed around the use of nonlinear analysis and how it helps to understand the main question of the manuscript. As it reads, the manuscript appears to use a variety of COP metrics without appropriate justification. This would strengthen the manuscript and help to draw the connections between motocross and postural control.

Authors: Thank you very much for your suggestion. This information has been expanded upon in the manuscript. See below:

Lines 35-45: “These tools allow the quantification of the motor behavior changes over time [19,20], giving information about the dynamic characteristics of sway patterns, which seem to be related to the ability to perform movement adjustments [23,24]. These measures have been applied to assess the COP dynamic, being the Detrending Fluctuations Analysis (DFA) and entropy the most widely used. Related, a higher complexity of COP fluctuations, as measured by DFA, has been related to the flexibility shown by younger women compared to older women in executing motor adjustments. (Cavanaugh et al., 2005; Roerdink et al., 2006; Wang & Yang, 2012). In addition, they have also been applied to relate postural balance to level in different sporting skills (Caballero et al., 2020; Caballer et al., 2021; Lamoth et al., 2009)”.

Lines 151-158: “Complexity refers to the structure of variability, i.e. how the fluctuations of the COP evolve over time (Caballero et al., 2014). Different methods of complexity can be ana-lyzed depending on the method used. FuzzyEn values indicate the degree of irregular-ity in a signal. This tool has been shown to be more consistent in relative terms, less dependent on data length, free parameter selection and more resistant to noise than other entropy measures (e.g. sample entropy) (Chen et al. 2009; Xie et al. 2010). This measure computes the repeatability of vectors of length m and m + 1 that repeat within a tolerance range of r of the standard deviation of the time series”.

Introduction

  1. Line 75-81: acyclic actions need to be further clarified and/or provide examples of what movements this related to. Some additional depth around functional specialization would help the reader better understand the concepts being addressed here.

Authors: Thank you very much for your suggestion. Some examples have been added in the manuscript. Please, see below:

Lines 83-90: “Similarly, it is not known whether BMX practice leads to interlimb balance asym-metries. Cyclic sports, such as swimming or running, did not show a marked asym-metry in the use of one leg or the other (Leinen et al., 2019; Samadi et al., 2009). Classic on-road cycling modalities are also considered symmetric sports (Maloney, 2019); however, BMX cyclists perform both cyclic and acyclic actions (e.g., static and dynamic balance stances, acrobatics, spins, jumps, etc.).”

  1. Line 88: What constitutes “better” balance performance? Please clarify.

Authors: Thank you very much for your suggestion. The concept has been clarified in the manuscript in lines 98-101.

“The sample of athletes will present better balance parameters than the control group, shown through lower COP dispersion and speed. In addition, it is not expected to find interlimb balance asymmetries due to the cyclic participation of both legs in this sport.”

Materials and methods

  1. Line 96: typo – raiders should be rider, correct?

Authors: Thank you very much for your correction. The typo has been changed in the manuscript, specifically you can find it in line 105.

  1. For the participants demographics - Some further insight for rider groups - years of experience, volume of racing over the past 6 months or so. A short description explaining the differences between freestyle and racing would be helpful for a reader not familiar with this population group.

Authors: Thank you very much for your suggestion. The BMX group was formed by international riders, it can be stated that the are high-experienced athletes. Some of this information has been added to the participants section in the manuscript. See lines 108-111: “The athlete group represented the BMX Spanish national team and competed in international tournaments, having 10 (±3) years of experience. They trained six days per week and had an annual competition volume of approximately 30-40 national and international competitions”.

In addition, a brief description about racing and freestyle BMX has also been added in lines 105-108: “of 7 freestyle and 12 racing riders (age: 21.9 ± 4.4 years, weight: 48.7 ± 23.6 kg, height: 170.6 ± 9.9 cm). Racing riders' objective is to be the fastest to complete a predefined circuit, while freestyle BMX is based on bike acrobatics (e.g., balancing, spins and jumps)”.

  1. Line 105: Was this conducted in a laboratory setting? Please clarify.

Authors: Thank you very much for your suggestion. It has been clarified in the manuscript in line 122-124: “Each participant performed four trials in a laboratory setting, two trials with the dominant leg and the other two with the non-dominant one”.

  1. Line 109: What determined the "best trial"? Please clarify.

Authors: Thank you very much for your suggestion. This sentence has been clarified in line 122: “considering this one the trials with lower bivariate variable error”.

  1. Line 112: “stand” should this be stance?

Authors: Thank you very much for your correction. The word “maintain” has been removed in order to correct the error in line 125.

  1. Data Analysis - It would be helpful to have background information around complexity within the introduction material or more in-depth explanations provided here in the methods. Not sure about the space limitations, but the Fuzzy ApEn approach definitely requires further details for readers as it is not a common metric used for COP analysis (it has been investigated and can be useful but the authors need further justification/rationale on its use.

Authors: Thank you very much for your suggestion. Further information about FuzzyEn is provided. Please, see below:

Lines 153-158: “FuzzyEn values indicate the degree of irregularity in a signal. This tool has been shown to be more consistent in relative terms, less dependent on data length, free pa-rameter selection and more resistant to noise than other entropy measures (e.g. sample entropy) (Chen et al. 2009; Xie et al. 2010). This measure computes the repeatability of vectors of length m and m + 1 that repeat within a tolerance range of r of the standard deviation of the time series.

Results

  1. Line 160: I did not see any reporting on the leg dominance for either group. This should be incorporated into some part of the results to help the reader understand the potential influence of leg dominance within the study. As a suggestion, Table 2 may be a good place to include this information.

Authors: Thank you very much for your suggestion. This information has been added in lines 208-211: “For the group control, the right leg was dominant for 95% of the participants (19/20). However, for only 40% of the sample (8/20) it was the best performing leg. While in the BMX group, the right leg was dominant for 89% of the participants (17/19), and for only 47% (9/19) it was the best performing leg”.

  1. Table 1: Table 1 needs to be moved to fit on a single page. Difficult to read when spread across two pages. The authors may consider removing the p-value column to simplify the presentation of the data given that they are using the * symbol to indicate significance - this would remove redundancies in the data presentation.

Authors: Thank you very much for your suggestion. The columns with the p-values have been removed from Table 1. See table in line 204.

  1. Table 2: Please clarify whether you are reporting mean and standard deviations.

Authors: Thank you very much for your suggestion. This information has been added to the description of the Table 2 in line 223.

  1. Line 195: “in the leg with lower COP dispersion” – can the authors please clarify this statement and whether this relates to leg/pedal dominance within the groups. As it reads, the authors selected out a leg independent of whether it was dominant or non-dominant.

Authors: Thank you very much for this suggestion. Changes in the text have been made to clarify this concept. See below:

Lines 227-231: “Considering that the participants did not reach the best performance in the balance task with their preferred leg, an additional comparison between groups was made according to the leg with lower COP dispersion (lower BVE). BMX athletes and the control group presented a similar performance in all variables since differences between groups were non-significant (Figure 2).

Discussion

  1. Line 223: “scattering variables” – please use consistent descriptors throughout the manuscript.

Authors: Thank you very much for this correction. The term “scattering” has been replaced by “dispersion” in line 261.

  1. Line 236-7: It is not clear how this study supports the first sentence of the paragraph. Further explanation around less regular and auto-correlated needed as well as the links to throwing velocity.

Authors: Thank you very much for this suggestion. Further information on this point has been provided. Please, see below:

Lines 274-280: “Previous studies have supported that individuals with more complex COP excursion, related to less regular (high entropy) and auto-correlated (low DFA) values, showed better performance in balance tasks (Caballero et al., 2020; Caballero et al., 2021). Caballero et al. (2020) even reported that handball players who presented a better balance and more complex COP excursion also exhibited higher accuracy and velocity in throwing, suggesting that greater variability in the movement would provide more resources to achieve better motor performance”.

  1. Line 249-50: A few new terms are introduced here that are not fully explained, particularly for novice readers. I think some further explanation/examples for bilateral asymmetric action and out-of-phase actions.

Authors: Thank you very much for this suggestion. The concept out-of-phase has been removed since the purpose of the sentence was to example symmetric and asymmetric actions, and not out-of-phase and in-phase symmetric actions. Please, see below the new clarification:

Lines 286-293: “These data agree with other studies that have reported the absence of asymmetry in balance between the dominant and non-dominant leg in cyclic and acyclic sports (Barone et al., 2010; Karadenizli et al., 2014; Leinen et al., 2019; Matsuda et al., 2008; Samadi et al., 2009). Although BMX riders perform both symmetric (i.e. pedaling) and asymmetric actions (i.e. jumping and balancing) (Maloney, 2019), it seems that symmetric actions prevail over asymmetric actions, since the experienced BMX riders measured in this study have shown similar performance in both extremities in a one-leg stance.”

Reviewer 2 Report

General comments:

The article compares the center of pressure (COP) of recreational athletes with elite BMX riders. Overall, the article is easy to read and well written, using linear and non-linear measures to characterize the COP. Laterality was taken into account since both lower limbs were considered for analysis. My main concern about this article concerns what is concluded, i.e., that "one-leg stance postural sway is not benefited by elite riders". The authors did not conduct a randomized controlled trial to assess whether they should have greater or lesser benefit relative to the motor skill in question. Also, when the “control” group is mentioned, we only know that they are recreational athletes. These athletes can also benefit from balance due to the training they do, which doesn't mean that BMX athletes don't also gain balance. This is the major limitation of this study. Statistically, I also have several concerns that will be explained in the specific comments. 

Specific comments:

Pag 2: What is the meaning of recreational athletes in the control group?

Pag 2-3, lines 96-99: replace “weight” by “body mass”, the units are “kg”.

Pag 3, lines 98-99: there are a quite difference between the body mass of the two groups and a different range in the standard deviation. How can the authors ensure that body mass is not a confounding variable since the statistical analysis did not consider this issue. 

Pag 4, lines 140-142: The reason for using FuzzyEn directly on postural sway directions is clearly obvious; however, the same cannot be said for DFA. All right, the authors intend to measure the complexity of this type of time series, but what information at the motor behavior level is supposed to be extracted? It is well established in the literature why this is done for stride intervals in gait and for RR intervals on the ECG. But in this case, what is expected to be extracted? The authors give a description regarding the time series itself, but do not report what is the relationship between the measurement and its physiological/motor meaning.

Pag 4, lines 161-164: At this part I got a little confused. The dependent variables to be analyzed are independent of each other, that is, they are separate analyses. So, we are not facing different levels that justify post-hoc analyses with corrections such as Holms or Bonferroni, or divisions of the alpha due to several ANOVAs being performed. In this case, we are dealing with dependent variables with differences between two groups. What should have been demonstrated is that the assumptions were guaranteed, such as normal distribution and homogeneity of variances in the independent t-test. The only levels that can be considered are the A-P and the M-L and the dominance, but in this case a mixed (A-P and M-L / dominance as within, groups as between) ANOVA should be performed. In the case of the correlations, what could be of interest it is partial correlation to understand the role of co-variables. If the options of dividing the alpha was due to the sample size, in other words, not enough to a mixed ANOVA analysis, this should be stated accordingly or more data should be collected (I know that the authors stated as limitation that it is not possible to enlarge sample size).

Pag 6, Figure 2: The authors used the “#” symbol to express significant differences with the control group, but the symbol does not appear in the Figure.

Discussion: 

a)     It is one thing to say that BMX athletes showed no differences in balance agility in the dominant or nation dominant limb, it is quite another to report that they do not benefit from increased balance ability. The control group are trained individuals who may do functional training or even be surfers. The activity of this control group will greatly influence the comparison results and the conclusions would be totally different. You must be very careful with this aspect. Should explain what kind of recreational athletes we are talking about.

b)    The authors stated that “Non-linear measurements of COP fluctuations have been implemented to explore the 221 dynamics of balance movements.” However, they used the term “nonlinear” and based the discussion on the correlations. A more detailed discussion of these variables should be provided, such as what was expected to be taken away (in the discussion it appears that the goal was to see whether they were correlated with the linear measures), and what the relationship of the values obtained are to others who have studied postural sway. 

c)     Thera are the limitation of control group characteristics.

My apologies for the delay in replying.

Author Response

Comments from Reviewer 2:

The authors would like to thank you for your advice and recommendations, which in our view have contributed to improve the quality of the paper. The authors’ responses to each comment (in bold) are presented below, and changes in the manuscript are presented in red.

General comments:

The article compares the center of pressure (COP) of recreational athletes with elite BMX riders. Overall, the article is easy to read and well written, using linear and non-linear measures to characterize the COP. Laterality was taken into account since both lower limbs were considered for analysis. My main concern about this article concerns what is concluded, i.e., that "one-leg stance postural sway is not benefited by elite riders". The authors did not conduct a randomized controlled trial to assess whether they should have greater or lesser benefit relative to the motor skill in question. Also, when the “control” group is mentioned, we only know that they are recreational athletes. These athletes can also benefit from balance due to the training they do, which doesn't mean that BMX athletes don't also gain balance. This is the major limitation of this study. Statistically, I also have several concerns that will be explained in the specific comments. 

Specific comments:

  1. Pag 2: What is the meaning of recreational athletes in the control group?

Authors: By recreational athletes the authors refer to people who practice a sport, in a non-professional way, other than BMX. In order to clarify this concept, his information has been added in lines 110-111: “physically active adults with no experience in BMX nor in sports involving balance in the performance (e.g. gymnastics, dance or surfing)”.

  1. Pag 2-3, lines 96-99: replace “weight” by “body mass”, the units are “kg”.

Authors: Thank you very much for this suggestion. The concept has been changed in the manuscript, in lines 106 and 114.

  1. Pag 3, lines 98-99: there are a quite difference between the body mass of the two groups and a different range in the standard deviation. How can the authors ensure that body mass is not a confounding variable since the statistical analysis did not consider this issue. 

Authors: Thank you very much for this observation. Thanks to this comment we have detected an error in the calculation of body mass. Now I there are hardly any differences in group means, and the standard deviation is smaller. So, we think that these small differences should not influence the results. Changes have been made to in line 106: “, body mass: 68.6 ± 12.1. kg”.

  1. Pag 4, lines 140-142: The reason for using FuzzyEn directly on postural sway directions is clearly obvious; however, the same cannot be said for DFA. All right, the authors intend to measure the complexity of this type of time series, but what information at the motor behavior level is supposed to be extracted? It is well established in the literature why this is done for stride intervals in gait and for RR intervals on the ECG. But in this case, what is expected to be extracted? The authors give a description regarding the time series itself, but do not report what is the relationship between the measurement and its physiological/motor meaning.

Authors: Thank you very much for your comment. It should be noted that, while the FuzzyEn measures the regularity of the time series, the DFA measures the auto-correlation of the time series. Both tools quantify the changes over time of the time series but, as some authors have previously indicated that entropy and autocorrelational measures do not provide the same information about the dynamic of the movement variability (Caballero et al, 2014; Stergiou & Decker, 2011). For this reason, these two measures are generally used as complementary measures. In order to clarify that, studies that used the DFA to analyze postural sway, specifically in relation to motor adjustments, have been added in the introduction section. Please, see below:

Lines 35-45: “These tools allow the quantification of the motor behavior changes over time (Barbado et al., 2012; Lamoth et al., 2009), giving information about the dynamic characteristics of sway patterns, which seem to be related to the ability to perform movement adjustments (Correl, 2008; Kilby et al., 2014). These measures have been applied to assess the COP dynamic, being the Detrending Fluctuations Analysis (DFA) and entropy the most widely used. High complexity of COP fluctuations measured by DFA has been related to the flexibility shown by younger women compared to older women in executing motor adjustments (Cavanaugh et al., 2005; Roerdink et al., 2006; Wang & Yang, 2012). In addition, it has also been applied to relate postural balance to performance in different sporting skills (Caballero et al., 2020; Caballero et al., 2021; Lamoth et al., 2009).”

Caballero C, Barbado D, Moreno FJ. Non-linear tools and methodological concerns measuring human movement variability: an overview. Eur J Hum Mov. 2014;32(0):61–81.

Stergiou N, Decker LM. Human movement variability, nonlinear dynamics, and pathology: Is there a connection? Hum Mov Sci. 2011;30(5):869–88.

  1. Pag 4, lines 161-164: At this part I got a little confused. The dependent variables to be analyzed are independent of each other, that is, they are separate analyses. So, we are not facing different levels that justify post-hoc analyses with corrections such as Holms or Bonferroni, or divisions of the alpha due to several ANOVAs being performed. In this case, we are dealing with dependent variables with differences between two groups. What should have been demonstrated is that the assumptions were guaranteed, such as normal distribution and homogeneity of variances in the independent t-test. The only levels that can be considered are the A-P and the M-L and the dominance, but in this case a mixed (A-P and M-L / dominance as within, groups as between) ANOVA should be performed. In the case of the correlations, what could be of interest it is partial correlation to understand the role of co-variables. If the options of dividing the alpha was due to the sample size, in other words, not enough to a mixed ANOVA analysis, this should be stated accordingly or more data should be collected (I know that the authors stated as limitation that it is not possible to enlarge sample size).

Authors: Thank you very much for your comment. The authors understand the reviewer's concern. As the reviewer indicates, some of the variables can be considered independent in the measurement of postural control. However, all of them measure parameters that could justify the fulfilment of the hypothesis since it mentions that there may be differences in postural control due to the practice of BMX sport. The fact of measuring several variables, which, although independent, all refer in one way or another to postural control, has led us to be more severe with the significance values in the comparisons and correlations. In addition, the authors understand that an ANOVA could have been used, but the small sample size and the fact that we used different leg dominance criteria led us to perform multiple t-tests to clarify the presentation of the results. The manuscript has been modified to clarify the statistical section. Please, see below.

Lines 187-190: “Because multiple balance parameters were used to assess postural control performance in the balance task, statistical significance was adjusted following Bonferroni criteria. Thus, statistical significance was set at p<0.002 for correlations and p<0.0016 for the t-test.”

  1. Pag 6, Figure 2: The authors used the “#” symbol to express significant differences with the control group, but the symbol does not appear in the Figure.

Authors: Thank you very much for this observation. As indicated in lines 230-232, there are no significant differences. For this reason, we have removed the sentence: "# Significant differences with control group (p< 0.05)", in Figure 2 caption, lines 237-237.

Discussion: 

  1. It is one thing to say that BMX athletes showed no differences in balance agility in the dominant or nation dominant limb, it is quite another to report that they do not benefit from increased balance ability. The control group are trained individuals who may do functional training or even be surfers. The activity of this control group will greatly influence the comparison results and the conclusions would be totally different. You must be very careful with this aspect. Should explain what kind of recreational athletes we are talking about.

Authors: Thank you very much for the suggestion. The authors agree and, in order to clarify it, the first part of the discussion has been rewritten. See below.

Lines 245-248: “It cannot be concluded that BMX training did not benefit riders for increased balance ability. However, the results suggest that the experience of training on the bicycle does not have a particular benefit on the one-leg balance performance compared to regular physical practice.”

In addition, as it has been indicated in previous comments, more information about the control group has been added. They were selected as they were physically active but without prior experience in BMX or in sports involved balance performance. Additional information regarding the control group description is given in the Participants section. See lines 111-114: “The control group consisted of 20 physically active adults with no experience in BMX nor in sports involving balance in the performance (e.g. gymnastics, dance or surfing) (age: 23.9 ± 3.6 years, weight: 68.4 ± 7.6 kg, height: 173.4 ± 6.9 cm).

  1. The authors stated that “Non-linear measurements of COP fluctuations have been implemented to explore the dynamics of balance movements.” However, they used the term “nonlinear” and based the discussion on the correlations. A more detailed discussion of these variables should be provided, such as what was expected to be taken away (in the discussion it appears that the goal was to see whether they were correlated with the linear measures), and what the relationship of the values obtained are to others who have studied postural sway. 

Authors: Thank you very much for the comment. The authors agree with the reviewer’s point of view but as the results did not reveal potential differences in COP fluctuation dynamics between groups, nor between legs, the discussion is therefore focused on the idea that it cannot be stated that BMX practice leads to better COP modulation through postural control adaptation strategies in a single-leg stance task. The correlational analysis showed that non-linear measurements would provide complementary information about the complexity of the postural sway, and other studies are mentioned related to how the analysis of COP complexity by nonlinear tools would help to reveal the true state of the postural balance control system (lines 264 to 272). For clarification, additional information has been added into de discussion section:

Lines 256-259: “The analysis did not reveal any potential difference between groups, nor between legs, in the non-linear variables. Therefore, it cannot be stated that BMX practice leads to better COP modulation through postural control adaptation strategies in a single-leg stance task. Additionally, it has been tried to improve the explanation in some studies mentioned in the discussion like in lines 274-280: “Previous studies have supported that individuals with more complex COP excursion, related to less regular (high entropy) and auto-correlated (low DFA) values, showed better performance in balance tasks (Caballero et al., 2020; Caballero et al., 2021). Caballero et al. (2020) even reported that handball players who presented a better balance and more complex COP excursion also exhibited higher accuracy and velocity in throwing, suggesting that greater variability in the movement would provide more resources to achieve better motor performance”

  1. There are the limitation of control group characteristics.

Authors: Thank you very much for your comment. The authors agree with the reviewer that the control group characteristics should be mentioned as a limitation. The following information has been added to the limitations paragraph in lines 320-325: “Secondly, regarding the control group, its characteristics cannot be disregarded as a possible confounding factor. Even though participants with experience in sports with high balance demands were excluded, the participants in the control group practiced different types of recreational sport, so the implications of their background in the single-leg stance tested in the present study cannot be precisely evaluated”. 

Reviewer 3 Report

An interesting paper, generally well written and with an interesting introduction, detailed data analysis with parametric and non-parametric measures. Some suggestions bellow in specific comments. The complexity of the balance test applied in adults, with non-health problems and physical active, seems with reduce capacity of distinguish elite and recreational athletes. This aspect should be deeply discuss on the paper.

Specific comments

Abstract

- Change “modalities” to “sports”.

Introduction

- Improve hypothesis writing, the use of the term mediated is not clear considering the sentence.

Material and Methods

L93-95 – the way sentence is written suggest that all, BMX riders and control group, are part of the BMX Spanish team.

L95-96 – “” seven riders in… and 12 raiders…” is duplicate information since is already mentioned on the previous sentence.

- Sample – It should be presented the differences in performance between groups – elite riders and recreational athletes. If one of the goals is to compare them, readers need to know how much they are different performing. By the way these recreational athletes are from BMX (other sports???). It lacks in information.

- Procedures in relation to the one-leg balance test – considering that one of the goals is to analyze possible difference between elite BMX riders and recreational athletes why choose a test with a medium level of difficulty? Why not to test removing the contribution of vision, with eyes closed, increasing the contribution of proprioception.

L127-128 – Could these 10 seconds of adaptation to show differences between groups. Would the group of accommodate trial initiation better?

L144-149 – It is not explicit for BMX riders how the second criteria for leg dominance were combined with the leg of kicking in the possible situation of them being different.

Results

L168-169 – Refer that Table 1 presents data from the total sample, riders and recreational athletes.

Discussion

Considering that the one-leg balance test is a standardize test, and that the discussion focus on the relation with studies in other sports, it would be interesting to discuss similarities or differences in the magnitude of the variables related to the COP.

Conclusion

In the abstract the CG is call recreational athletes but, in the conclusion, they are mentioned as physically active people. Also, in methods section nothing is mentioned to describe this group. The reading of the abstract suggest that the CG are also athletes of recreational level of BMX…Clearly define the group and if possible present some data/information that explain their level of performance.

Author Response

Comments from Reviewer 3:

The authors would like to thank you for your advice and recommendations, which in our view have contributed to improve the quality of the paper. The authors’ responses to each comment (in bold) are presented below, and changes in the manuscript are presented in red.

An interesting paper, generally well written and with an interesting introduction, detailed data analysis with parametric and non-parametric measures. Some suggestions bellow in specific comments. The complexity of the balance test applied in adults, with non-health problems and physical active, seems with reduce capacity of distinguish elite and recreational athletes. This aspect should be deeply discuss on the paper.

Abstract

  1. Change “modalities” to “sports”.

Authors: Thank you very much for this suggestion. The concept has been changed in the manuscript, in line 9.

Introduction

  1. Improve hypothesis writing, the use of the term mediated is not clear considering the sentence.

Authors: Thank you very much for this suggestion. The whole sentence has been rephrased for clarification. Please, see lines 98-101: “It is hypothesized that the sample of athletes will present better balance parameters than the control group, shown through lower COP dispersion and speed. In addition, it is not expected to find interlimb balance asymmetries due to the cyclic participation of both legs in this sport”.

Material and Methods

  1. L93-95 – the way sentence is written suggest that all, BMX riders and control group, are part of the BMX Spanish team.

Authors: Thank you very much for this suggestion. These sentences have been modified in the manuscript in lines 105-106: “7 freestyle and 12 racing riders (age: 21.9 ± 4.4 years, weight: 48.7 ± 23.6 kg, height: 170.6 ± 9.9 cm).”

  1. L95-96 – “” seven riders in… and 12 raiders…” is duplicate information since is already mentioned on the previous sentence.

Authors: Thank you very much for this correction. The duplicate information has been fixed in lines 105-107. The reviewer can check it in the previous comment.

  1. Sample – It should be presented the differences in performance between groups – elite riders and recreational athletes. If one of the goals is to compare them, readers need to know how much they are different performing. By the way these recreational athletes are from BMX (other sports???). It lacks in information.

Authors: Thank you very much for this suggestion. It must be pointed out that none of the control group were BMX riders or similar. They were involved in recreational sports; therefore, no comparison of experience or performance can be made between the BMX group. This information has been expanded in in the manuscript. See below:

Lines 105-114: “The BMX group was made up of 7 freestyle and 12 racing riders (age: 21.9 ± 4.4 years, weight: 68.6 ± 12.1. kg,  height: 170.6 ± 9.9 cm). Racing riders' objective is to be the fastest one to complete a predefined circuit, while freestyle BMX is based on bike acrobatics (e.g., balancing, spins, jumps, etc.). The athlete group represented the BMX Spanish national team and competed in international tournaments. The control group consisted of 20 physically active adults with no experience in BMX (age: 23.9 ± 3.6 years, weight: 68.4 ± 7.6 kg, height: 173.4 ± 6.9 cm)”.

  1. Procedures in relation to the one-leg balance test – considering that one of the goals is to analyze possible difference between elite BMX riders and recreational athletes why choose a test with a medium level of difficulty? Why not to test removing the contribution of vision, with eyes closed, increasing the contribution of proprioception.

Authors: Thank you very much for this suggestion. The reviewer underlines a good point in this suggestion. This idea has been added as a limitation of this research at the end of the discussion. See lines 326-332: “In addition, the one-leg static balance test used in the present study may not have been very demanding for any of the groups, so that test difficulty may be considered as limitation in the findings. However, this test was chosen for the comparison between groups because both the BMX athletes and the control group can easily execute it and to check if a general balance test can provide useful information. It is encouraged to use more complex tests in future research (e.g., open vs closed eyes one-leg stances, dynamic vs static balance tests, etc.)”.

  1. L127-128 – Could these 10 seconds of adaptation to show differences between groups. Would the group of accommodate trial initiation better?

Authors: Thank you very much for your question. As it has been explained in the Data Analysis and Reduction, this first 10 s of each trial were removed because they can cause non-stationarity in the CoP timeseries (van Dieën et al., 2010). This non-stationarity affects the reliability and robustness of the nonlinear measures (Peng et al., 2009; Caballero et al., 2013), Thus, in order to properly use this type of tools, it must be avoided.

van Dieën, J. H., Koppes, L. L. J., & Twisk, J. W. R. (2010). Postural sway parameters in seated balancing; their reliability and relationship with balancing performance. Gait & Posture, 31(1), 42–46. https://doi.org/10.1016/j.gaitpost.2009.08.242

Peng, C. K., Costa, M., and Goldberger, A. L. (2009). Adaptive data analysis of complex fluctuations in physiologic time series. Adv. Adapt. Data Anal. 1 (1), 61–70. doi:10.1142/S1793536909000035

Caballero, C., Barbado, D., and Moreno, F. J. (2013). El procesado del desplazamiento del centro de presiones para el estudio de la relación complejidad/rendimiento observada en el control postural en bipedestación. Med. del deporte 6 (3), 101–107.

  1. L144-149 – It is not explicit for BMX riders how the second criteria for leg dominance were combined with the leg of kicking in the possible situation of them being different.

Authors: Thank you very much for your suggestion. Additional information has been added to clarify the rationale of using different leg dominance criteria. Please, see lines 168-169: “Different leg dominance criteria were followed in order to discuss the most sensitive one when identifying between-groups and between-legs performance”.

However, it should be noted that statistical analysis comparing the three leg dominance criteria was not carried out because our aim was to discuss the results (athlete’s performance compared to controls, and interlimb asymmetries) following different criteria (e.g., the well-known kicking leg criteria, a specific criterion for BMX, or a performance criterion), without going deeper in differences between criteria.

Results

  1. L168-169 – Refer that Table 1 presents data from the total sample, riders and recreational athletes.

Authors: Thank you very much for your suggestion. The following text has been added to Table 1 (line 204) description: “(this analysis included BMX athletes and controls; n = 39)”.

Discussion

  1. Considering that the one-leg balance test is a standardize test, and that the discussion focus on the relation with studies in other sports, it would be interesting to discuss similarities or differences in the magnitude of the variables related to the COP.

Authors: Thank you very much for your suggestion. As the reviewer suggested, the one-leg balance test has been widely used in athletes as well as healthy and injured adults. However, these studies frequently differ in their procedures or variables analyzed (e.g., maximum balance time, COP length, COP area, etc.). For this reason, a control group was added to the study design in order to focus the discussion on interlimb and intergroup comparison within the present study.

Conclusion

  1. In the abstract the CG is call recreational athletes but, in the conclusion, they are mentioned as physically active people. Also, in methods section nothing is mentioned to describe this group. The reading of the abstract suggest that the CG are also athletes of recreational level of BMX…Clearly define the group and if possible present some data/information that explain their level of performance.

Authors: The CG consisted of physically active people (not BMX recreational athletes). This information has been amended in the abstract, introduction and methods, and additional information regarding CG description is given in the Participants section. See lines 111-114: “The control group consisted of 20 physically active adults with no experience in BMX nor in sports involving balance in the performance (e.g. gymnastics, dance or surfing) (age: 23.9 ± 3.6 years, weight: 68.4 ± 7.6 kg, height: 173.4 ± 6.9 cm).

Round 2

Reviewer 3 Report

In my opinion the authors improved the paper having in consideration my concerns. I have no further suggestions.